# High diversity of mosquito vectors in Cambodian primary schools and consequences for arbovirus transmission

Sebastien Boyer[1]*, Sebastien Marcombe[2], Sony Yean[1], Didier Fontenille[1]

1 Medical and Veterinary Entomology Unit, Institut Pasteur du Cambodge, Boulevard Monivong, Phnom Penh, Cambodia, 2 Medical Entomology Unit, Ministry of Health, Institut Pasteur du Laos, Vientiane, Lao PDR

* sboyer@pasteur-kh.org

**Data Availability Statement:** All relevant data are within the paper and its Supporting Information files.

## Abstract

Only few data exist in Cambodia on mosquito diversity and their potential role as vectors. Many arboviruses, such as dengue and Japanese encephalitis, are endemic and mostly affect children in the country. This research sets out to evaluate vector relative abundance and diversity in primary schools in Cambodia in an attempt to explain the apparent burden of dengue fever, severe dengue (DEN), Japanese encephalitis (JE), other arboviral diseases and malaria among children, 15 years and under, attending selected primary schools through vector surveys. Entomological surveys were implemented in primary schools in two provinces of Cambodia to assess the potential risk of exposure of schoolchildren to mosquito vector species. Light traps and BG traps were used to collect adult mosquitoes in 24 schools during the rainy and dry seasons of 2017 and 2018 in Kampong Cham and Tboung Khmum provinces. A total of 61 species were described, including *Aedes*, *Culex* and *Anopheles* species. The relative abundance and biodiversity of mosquito species were dependent on the month and school. Of the 37,725 mosquitoes caught during the study, three species accounted for three-quarters of the relative abundance: *Culex vishnui*, *Anopheles indefinitus* and *Culex quinquefasciatus*. More importantly, nearly 90% of the mosquitoes caught in the schools were identified as potential vectors of pathogens including Japanese encephalitis, dengue, and malaria parasites. Our results showed that schools in Cambodia represent a risk for vector-borne disease transmission and highlight the importance of implementing vector control in schools in Cambodia to decrease the risk of transmission.

## Introduction

Mosquito-borne pathogens such as arboviruses and malaria parasites are transmitted through a high diversity of species belonging to three genera: *Aedes*, *Anopheles*, and *Culex*. Worldwide, *Anopheles* mosquitoes are the main vectors of malaria and despite an undeniable improvement of the situation [1], there was still an estimated 219 million cases and 435,000 death worldwide for the year 2017 (WHO [2], malaria report). The genera *Aedes* and *Culex* mainly transmit

**Funding:** The study was supported by ECOMORE 2 project, coordinated by Institut Pasteur and financially supported by AFD (Agence Française pour le Développement). The funders had no role in study design, data collection and analysis, decision to publish, or preparation of the manuscript.

**Competing interests:** The authors have declared that no competing interests exist.

arboviruses responsible for diseases in humans such as dengue fever, yellow fever, West Nile fever, Rift Valley fever, Zika, Chikungunya, and Japanese encephalitis. In recent years, an increase in the burden related to arboviruses has been reported worldwide [3].

In Cambodia, dengue, Chikungunya, Zika, and Japanese encephalitis viruses circulate [4]. Dengue is probably the most important of these diseases and since the massive dengue epidemic in 1995 with more than 400 deaths, the number of cases has been monitored [4]. Major dengue epidemics occurred in 2007 with 39,618 cases (396 deaths) and 2012 with 42,362 cases (189 deaths); the estimated incidence among children less than 7 years-old of age is 41.1/1,000 person-seasons [5] with an incidence of dengue haemorrhagic fever and dengue shock syndrome of 4.1 per 1,000 children under 15 years old [6; 7]. During 2018, the Cambodian capital Phnom Penh faced a high circulation of dengue cases with 9,445 cases including 6 deaths reported by the National Dengue Control Program (Ministry of Health, Phnom Penh, Cambodia). Japanese encephalitis virus is also endemic in Cambodia with an incidence estimated at 0.1 case per 1,000 children under 15 years old [8; 9]. In 2017, malaria incidence was estimated to be between 186,000 and 236,000 cases per year [10].

There are 43 genera and 3,530 mosquito species currently described worldwide [11] and the presence of 20 genera and 243 species is estimated in Cambodia. This estimate is mainly based on institutional collections (Institut de Recherche pour le Développement, Montpellier, France, and Smithsonian Institute, Washington D.C., United States) and on studies implemented before the Khmer rouge period (1975–1979). Except studies specifically targeting dengue and malaria vectors [12, 13, 14], there is no comprehensive study on the biodiversity of other mosquitoes of medical importance in Cambodia. Specifically, there is a lack of knowledge on the Cambodian mosquito diversity potentially present in the school premises. Indeed, the classical infrastructure of school buildings, with open ventilation in the top of the walls allowing mosquitoes to enter the premises to access shade and cooler areas to rest but also to have a potential blood meal.

In this study, schools were chosen to carry out a first inventory based on the fact that schools are areas of high density of children in environments at risk of vector-borne pathogen transmission. In general, children of school attending age are from 4 to 15 years old, are more naive and sensitive to arboviruses and parasites and therefore are at risk to develop associated diseases. The objective of this study was to determine the mosquito species diversity and relative abundance in schools from two provinces of Cambodia known for the circulation of several dengue serotypes, and with potential risk of other arboviruses and parasites.

## Materials and methods

### Study area

Table 1 shows the characteristics of the schools selected for the study and their GPS coordinates. A total of 24 primary schools located in rural, peri-urban and urban areas were chosen randomly in Kampong Cham and Tboung Khmum provinces (Fig 1). The National Ethics Committee for Health Research of the Ministry of Health of the Kingdom of Cambodia approved the field site access and the permit N° 113NECHR was obtained on 02 May 2017. The study was also fully supported by the National Center for Parasitology, Entomology and Malaria Control, Ministry of Health (Official mail N° 348/17 dated from 28 April 2017), and by the Ministry of Education, Youth and Sport (Official mail N° 2592 dated from 27 April 2017). Mosquitoes were collected on 4 different dates, every 3 months during both dry and rainy seasons: (1) 5–19 May 2017, (2) 9–17 August 2017, (3) 10–18 November 2017, and (4) 2–14 February 2018. Typically, the month of May represents the end of the dry season, August the rainy season, November the end of the rainy season, and February the dry season.

**Table 1. Descriptive data of the 24 schools in Kampong Cham and Tboung Khmum provinces, Cambodia.**

| Schools | Development | Pagoda | School area (m²) | Number of students | Number of classrooms | Classrooms with traps | BG traps | CDC Light Traps | Latitude | Longitude |
|---|---|---|---|---|---|---|---|---|---|---|
| Rumchek | Peri-urban | yes | 3692 | 496 | 12 | 6 | 3 | 3 | 12˚06.036' | 105˚31.062' |
| Ro-ang Leu | Peri-urban | yes | 7466 | 285 | 7 | 6 | 3 | 3 | 12˚03.183' | 105˚29.187' |
| Lvea | Peri-urban | yes | 14191 | 694 | 17 | 6 | 3 | 3 | 12˚20.401' | 105˚17.024' |
| Sre Praing | Peri-urban | no | 4951 | 383 | 10 | 6 | 3 | 3 | 12˚23.249' | 105˚12.672' |
| Prek Kak | Peri-urban | no | 9229 | 568 | 13 | 6 | 3 | 3 | 12˚13.466' | 105˚31.845' |
| Hann Chey | Peri-urban | no | 21550 | 780 | 18 | 6 | 3 | 3 | 12˚08.145' | 105˚31.449' |
| Wat Thmei | Rural | no | 5051 | 578 | 14 | 6 | 3 | 3 | 12˚04.262' | 105˚25.773' |
| Paprak | Rural | yes | 14957 | 393 | 11 | 6 | 3 | 3 | 12˚15.617' | 105˚21.769' |
| Trapaing Russey | Rural | no | 8360 | 449 | 11 | 6 | 3 | 3 | 12˚19.956' | 105˚22.427' |
| Chamka Andaung | Rural | no | 4467 | 335 | 10 | 6 | 3 | 3 | 12˚20.414' | 105˚10.673' |
| Sre Preal | Rural | no | 9768 | 234 | 10 | 4 | 2 | 2 | 12˚18.445' | 105˚16.352' |
| Kbal Damrei | Rural | no | 2811 | 211 | 6 | 3 | 1 | 2 | 12˚07.521' | 105˚08.771' |
| Chheu Bak | Rural | no | 6804 | 209 | 7 | 4 | 2 | 2 | 12˚11.622' | 105˚09.788' |
| O Ta Thok | Rural | yes | 2500 | 191 | 6 | 4 | 2 | 2 | 12˚10.403' | 105˚13.721' |
| Svay Areak | Rural | no | 1857 | 223 | 6 | 4 | 2 | 2 | 12˚05.945' | 105˚08.208' |
| Svay Prey | Rural | yes | 4280 | 216 | 6 | 6 | 3 | 3 | 12˚01.860' | 105˚07.690' |
| Koh Pen | Rural | no | 3445 | 479 | 14 | 6 | 3 | 3 | 11˚57.313' | 105˚27.299' |
| Khvet Thom | Urban | no | 12251 | 252 | 12 | 6 | 3 | 3 | 12˚03.526' | 105˚13.304' |
| Angkor | Urban | no | 9829 | 1112 | 25 | 6 | 3 | 3 | 12˚00.057' | 105˚26.560' |
| AngKor Chey | Peri-urban | no | 7090 | 563 | 11 | 6 | 3 | 3 | 11˚53.140' | 105˚44.018' |
| Toul Vihear | Peri-urban | no | 10047 | 584 | 14 | 6 | 3 | 3 | 11˚57.452' | 105˚31.877' |
| Punley | Rural | yes | 20586 | 453 | 12 | 6 | 3 | 3 | 11˚49.811' | 105˚43.847' |
| Steung Penh | Rural | no | 8292 | 175 | 6 | 4 | 2 | 2 | 11˚52.485' | 105˚35.500' |
| Samrong Borei | Rural | no | 6486 | 315 | 10 | 6 | 3 | 3 | 11˚53.066' | 105˚38.275' |

## Adult mosquito collection and identification

Sixty-five BG-sentinel traps® (Biogents, Regensburg, Germany) and 66 CDC light traps® (JWHock, Gainsville, USA) were used to collect adult mosquitoes. BG-sentinel traps were used with a lure designed by BioGents ® Company consisting of a combination of substances such as lactic acid, ammonia, and fatty acids that can be found on human skin [15]. Traps were set up during 24 hours inside classrooms with 1 trap per classroom. Following the traditional Cambodian construction, all classes of all schools have openings in the tops of each wall to let the air circulates and reduce the high temperatures occurring during the daytime, allowing the free circulation of mosquitoes. Depending on the size of the schools and the number of classrooms, three (or less) BG traps and three (or less) CDC light traps were used with a limit of a total of six traps per school (Table 1). After each collection, mosquitoes were conserved in an electric icebox at +4˚C and first identified in the field the same day. Then, samples were brought back to the laboratory, still at +4˚C, in Phnom Penh for a second identification. In the field and in the laboratory, mosquitoes were identified under stereomicroscope and microscope using morphological mosquito identification keys from Southeast Asia countries [16; 17].

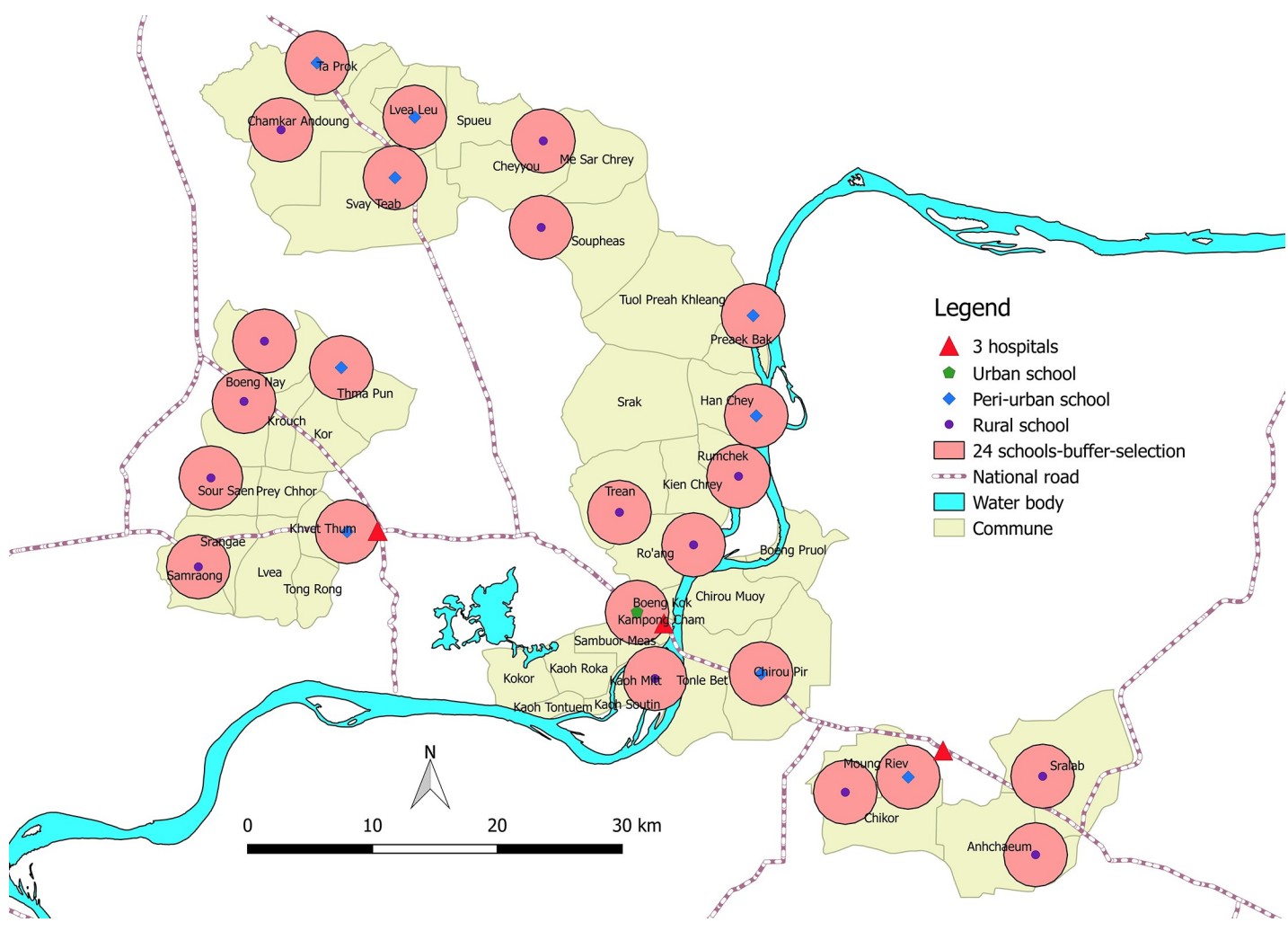

**Fig 1. Map representing the spatial distribution of the 24 schools sampled in Kampong Cham and Tboung Khmum provinces, Cambodia.**

### Statistical analysis

The association of the different parameters characterizing the schools and the period of collection, i.e. month, school, urbanization, presence of pagoda, school area, the number of classroom and children in the school and the relative densities of mosquito populations (i.e. all mosquitoes, all vector mosquitoes, Japanese encephalitis virus vectors, Dengue virus vectors, malaria plasmodium vectors) in the considered schools, were tested by a generalized linear mixed model using Template Model Builder with the assignation of 'school' as random variables, in order to determine the main significant factors. After verification of residual normality (Shapiro-test) and correction of the surdispersion, the results of the model and test were obtained. A supplementary analysis was implemented to represent the month effect on dengue vectors (Kruskal-Wallis test).

## Results

### Relative abundance and diversity of mosquito species

The list of the genera and mosquito taxa collected during the four collection sessions in the 24 schools is presented in Table 2. A total of 35,725 mosquitoes belonging to at least 55 mosquito

**Table 2.** Mosquito species caught during 4 trapping sessions in Kampong Cham and Tboung Khmum provinces, Cambodia, in May, August and November 2017, and February 2018.

| Mosquito species | May | August | November | February | Total | (%) | Potential Vectors |
|---|---|---|---|---|---|---|---|
| Aedeomyia (1) | | | | | | | |
| *Aedeomyia catasticta* | 34 | 8 | 8 | 151 | 201 | 0.56% | |
| Aedes (5) | | | | | | | |
| *Aedes aegypti* | 389 | 164 | 33 | 188 | 774 | 2.17% | DENV, ZIKV, CHIKV, JEV, RVFV, WNV, YFV, RRV |
| *Aedes albopictus* | 139 | 162 | 54 | 40 | 395 | 1.11% | DENV, ZIKV, CHIKV, JEV, RVFV, WNV, YFV, RRV |
| *Aedes lineatopennis* | 2 | 1 | 1 | 14 | 18 | 0.05% | |
| *Aedes malayensis* | 0 | 2 | 0 | 0 | 2 | 0.01% | DENV |
| *Aedes vexans* | 2 | 1 | 1 | 2 | 6 | 0.02% | ZIKAV, JEV, RVFV |
| unidentified *Aedes* | 83 | 29 | 25 | 61 | 198 | 0.55% | |
| Anopheles (19) | | | | | | | |
| *Anopheles annularis* | 0 | 2 | 1 | 0 | 3 | 0.01% | |
| *Anopheles argyprous* | 0 | 2 | 0 | 0 | 2 | 0.01% | |
| *Anopheles barbirostris* | 1 | 3 | 6 | 11 | 21 | 0.06% | MAL |
| *Anopheles barbumbrosus* | 10 | 1 | 49 | 21 | 81 | 0.23% | MAL |
| *Anopheles campestris* | 2 | 22 | 2 | 5 | 31 | 0.09% | MAL |
| *Anopheles crawfordi* | 0 | 0 | 6 | 0 | 6 | 0.02% | |
| *Anopheles hodgkini* | 0 | 4 | 0 | 0 | 4 | 0.01% | |
| *Anopheles indefinitus* | 1543 | 842 | 971 | 1258 | 4614 | 12.92% | |
| *Anopheles nitidus* | 3 | 45 | 1 | 4 | 53 | 0.15% | |
| *Anopheles peditaeniatus* | 19 | 332 | 280 | 156 | 787 | 2.20% | |
| *Anopheles phillippinensis* | 0 | 9 | 0 | 0 | 9 | 0.03% | MAL |
| *Anopheles separatus* | 0 | 2 | 0 | 0 | 2 | 0.01% | |
| *Anopheles sinensis* | 2 | 21 | 37 | 46 | 106 | 0.30% | MAL |
| *Anopheles subticus* | 4 | 1 | 0 | 0 | 5 | 0.01% | |
| *Anopheles tessellatus* | 3 | 10 | 0 | 0 | 13 | 0.04% | |
| *Anopheles vagus* | 50 | 27 | 0 | 0 | 77 | 0.22% | MAL, JEV |
| unidentified *Anopheles* | 114 | 53 | 529 | 133 | 829 | 2.32% | |
| Armigeres (1) | | | | | | | |
| *Armigeres subalbatus* | 185 | 108 | 53 | 43 | 389 | 1.09% | DENV, JEV |
| *Armigeres sp* | 1 | 1 | 2 | 22 | 26 | 0.07% | |
| Coquilletidia (2) | | | | | | | |
| *Coquillettidia crassipes* | 56 | 27 | 3 | 8 | 94 | 0.26% | |
| *Coquillettidia ochracea* | 4 | 23 | 3 | 5 | 35 | 0.10% | |
| *Coquillettidia sp1* | 1 | 7 | 2 | 1 | 11 | 0.03% | |
| Culex (15) | | | | | | | |
| *Culex bitaeniorhynchus* | 33 | 324 | 25 | 48 | 430 | 1.20% | JEV, RVFV |
| *Culex brevipalpis* | 678 | 360 | 146 | 73 | 1257 | 3.52% | |
| *Culex fuscocephala* | 35 | 106 | 90 | 354 | 585 | 1.64% | JEV |
| *Culex gelidus* | 27 | 228 | 110 | 546 | 911 | 2.55% | JEV, RRV |
| *Culex hutchinsoni* | 0 | 0 | 0 | 8 | 8 | 0.02% | |
| *Culex malayi* | 8 | 9 | 0 | 1 | 18 | 0.05% | |
| *Culex nigropunctatus* | 4 | 59 | 29 | 12 | 104 | 0.29% | |
| *Culex quinquefasciatus* | 1215 | 745 | 314 | 702 | 2976 | 8.33% | ZIKAV, JEV, RVFV, WNV, RRV |
| *Culex sinensis* | 0 | 28 | 0 | 0 | 28 | 0.08% | |
| *Culex sitiens* | 0 | 5 | 0 | 0 | 5 | 0.01% | |
| *Culex tritaeniorhynchus* | 150 | 320 | 49 | 33 | 552 | 1.55% | JEV, RVFV, WNV |

*(Continued)*

**Table 2.** (Continued)

| Mosquito species | May | August | November | February | Total | (%) | Potential Vectors |
|---|---|---|---|---|---|---|---|
| *Culex vishnui.g* | 1708 | 5670 | 5243 | 5801 | 18422 | 51.57% | JEV |
| *Culex whitmorei* | 0 | 12 | 0 | 0 | 12 | 0.03% | |
| *Culex sp1* | 24 | 5 | 4 | 2 | 35 | 0.10% | |
| *Culex sp2* | 2 | 32 | 21 | 4 | 59 | 0.17% | |
| *Culex sp3* | 0 | 0 | 0 | 4 | 4 | 0.01% | |
| *unidentified Culex* | 46 | 96 | 24 | 37 | 203 | 0.57% | |
| Lutzia (2) | | | | | | | |
| *Lutzia fuscana* | 3 | 5 | 0 | 60 | 68 | 0.19% | |
| *Lutzia halifaxii* | 0 | 1 | 0 | 0 | 1 | 0.00% | |
| *Lutzia vorax* | 0 | 1 | 0 | 0 | 1 | 0.00% | |
| *Lutzia sp* | 0 | 1 | 0 | 0 | 1 | 0.00% | |
| Mansonia (2) | | | | | | | |
| *Mansonia annulifera* | 5 | 45 | 5 | 0 | 55 | 0.15% | |
| *Mansonia uniformis* | 325 | 360 | 180 | 65 | 930 | 2.60% | RVFV, WNV, RRV |
| *Mansonia sp* | 1 | 0 | 2 | 1 | 4 | 0.01% | |
| Mimomyia (3) | | | | | | | |
| *Mimoyia aurea* | 0 | 0 | 1 | 0 | 1 | 0.00% | |
| *Mimomyia elegans* | 6 | 5 | 1 | 5 | 17 | 0.05% | |
| *Mimomyia hybrida* | 5 | 9 | 0 | 0 | 14 | 0.04% | |
| *Mimomyia luzonensis* | 4 | 26 | 13 | 61 | 104 | 0.29% | |
| *Mimomyia sp* | 9 | 15 | 2 | 4 | 30 | 0.08% | |
| Uranotaenia (8) | | | | | | | |
| *Uranotaenia micans* | 0 | 0 | 1 | 0 | 1 | 0.00% | |
| *Uranotaenia nivipleura* | 0 | 0 | 4 | 2 | 6 | 0.02% | |
| *Uranotaenia rampae* | 29 | 11 | 3 | 1 | 44 | 0.12% | |
| *Uranotaenia subnormalis/latelaris* | 3 | 0 | 1 | 2 | 6 | 0.02% | |
| *unidentified Uranotaenia* | 3 | 3 | 25 | 8 | 39 | 0.11% | |
| Unidentified specimen | 0 | 0 | 2 | 0 | 2 | 0.01% | |
| Total | **6970** | **10390** | **8362** | **10003** | **35725** | | |

DENV = Dengue virus, ZIKV = Zika virus, CHIKV = Chikungunya virus, JEV = Japanese Encephalitis virus, RVFV = Rift Valley Fever virus, WNV = West Nile Virus, YFV = Yellow Fever virus, RRV = Ross River virus, MAL = *Plasmodium* spp.

species were collected. They were composed of 10 genera: *Aedeomyia* (1 species), *Aedes* (4), *Anopheles* (18), *Armigeres* (1), *Coquillettidia* (3) *Culex* (16), *Lutzia* (3), *Mansonia* (2), *Mimo-myia* (4), *Uranotaenia* (4). On average, 25 ± 4 mosquito species were found in all the schools, representing an important biodiversity of species compared to the 243 number of species already described in Cambodia. The genera *Aedes*, *Anopheles*, *Culex* and, *Mansonia* were collected in the 24 schools, and there were at least 8 different genera in 21 out of 24 schools. The genus *Culex* represented 71.7% of the total mosquitoes collected (25,609 mosquitoes/35,725) with 16 species identified. Sixteen *Anopheles* species (18.6%; 6,643/35,725), and 5 *Aedes* species (3.9%; 1,393/35,725) were collected (Table 2). The 3 most abundant species were *Cx. vishnui* (18,422 mosquitoes, 51.6%), *An. indefinitus* (4,614; 12.9%) and *Cx. quinquefasciatus* (2,976; 8.3%), representing together 72.8% of all the mosquitoes caught during the 4 collections. These 3 species were present in all the 24 schools. Nine other species were also collected in all the schools: *Ae. aegypti*, *Ae. albopictus*, *An. peditaeniatus*, *Cx. bitaeniorhynchus*, *Cx. brevipalpis*, *Cx. fuscocephala*, *Cx. gelidus*, *Cx. tritaeniorhynchus* and *Ms. uniformis*. Schools with the largest

number of species were Sre paing (37 species), Paprak (32) and Khvet Thom (30). The 2 schools with the least mosquito species were Ta Thok and Svay Areak, with 16 species each: they were the only two schools with less than 21 mosquito species. In August 2017, during the rainy season, we recorded the largest relative abundance of mosquitoes (10,196) with the highest number of species (48 species), while at the end of the dry season, May 2017, yielded the lowest relative abundance (6,713; 38 species). During the November (2017) and February (2018) collection sessions, the lowest number of mosquito species (35) was collected.

## Mosquito vector species and pathogen transmission risk

Of all the 61 mosquito species collected, at least 17 species are considered as potential vectors of pathogens (Table 2): *Ae. aegypti*, *Ae. albopictus*, *Ae. vexans*, *An. barbirostris*, *An. indefinitus*, *An. vagus*, *Armigeres subalbatus*, *Cx. bitaeniorhynchus*, *Cx. fuscocephala*, *Cx. gelidus*, *Cx. quinquefasciatus*, *Cx. sitiens*, *Cx. tritaeniorhynchus*, *Cx. vishnui*, *Mansonia uniformis* and *Tripteroides powelli*. Among these species, we identified vectors of *Plasmodium* spp. (0.93%), dengue virus (4%), Zika virus (12%), Chikungunya virus (3%), Japanese Encephalitis virus (71%), Rift Valley Fever virus (17%), West Nile Fever virus (16%), Yellow fever virus (3%), and Ross River virus (17%). The 17 potential vector species represented 86.5% (25,206/29,155) of all the mosquitoes collected during the 4 entomological missions. The proportion was 77%, 87%, 85% and 91% in May, August, November and February, respectively (Table 2).

The generalized linear mixed model with random-school effect showed only a month effect. For all vectors and JEV vectors, there are significantly less mosquitoes in May, that represents the driest month, and the last one before the rainy season (S1 Table in S1 File). All other factors characterizing the school environment were not significant.

Only the DENV vector abundance was significantly different for all the months. There were significantly more DENV vectors in schools in November (end of the rainy season) with 29.7 vectors per school and less in February (dry season) with 5.9 (Fig 2). In May and August, respectively, the average was 18.1 and 11.3 DENV vectors per school. The interaction of the effects of months and schools for DENV vectors' relative abundance is represented on Fig 3. The important relative abundance of DENV vectors in May is observed within all the schools, especially in Ta Prok and Soupheas schools (see Fig 1 for name schools). The more constant relative abundance of DENV along the year is observed along the Mekong River.

The maps representing the distribution of all mosquitoes, all vectors, JEV vectors and *Plasmodium* sp. vectors highlight the significant distribution of the different vectors regarding the different schools (Fig 4). As an illustration, JEV vectors are present in important relative

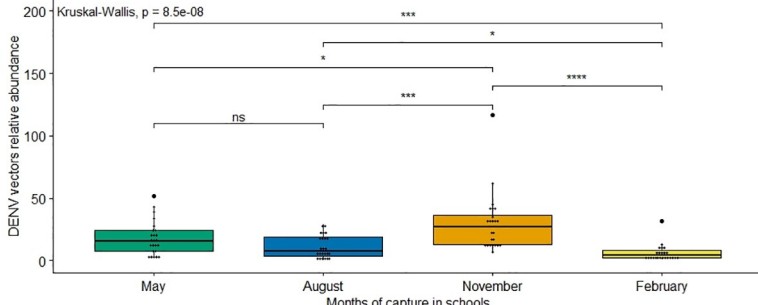

**Fig 2. Relative abundance of Dengue virus vectors caught in 24 schools during 4 different months.** Average comparison of the number of mosquitoes / night / school was tested with a Kruskal-Wallis tests (ns meaning non significant with p > 0.05; * p < 0.05; ** p < 0.01; *** p < 0.001).

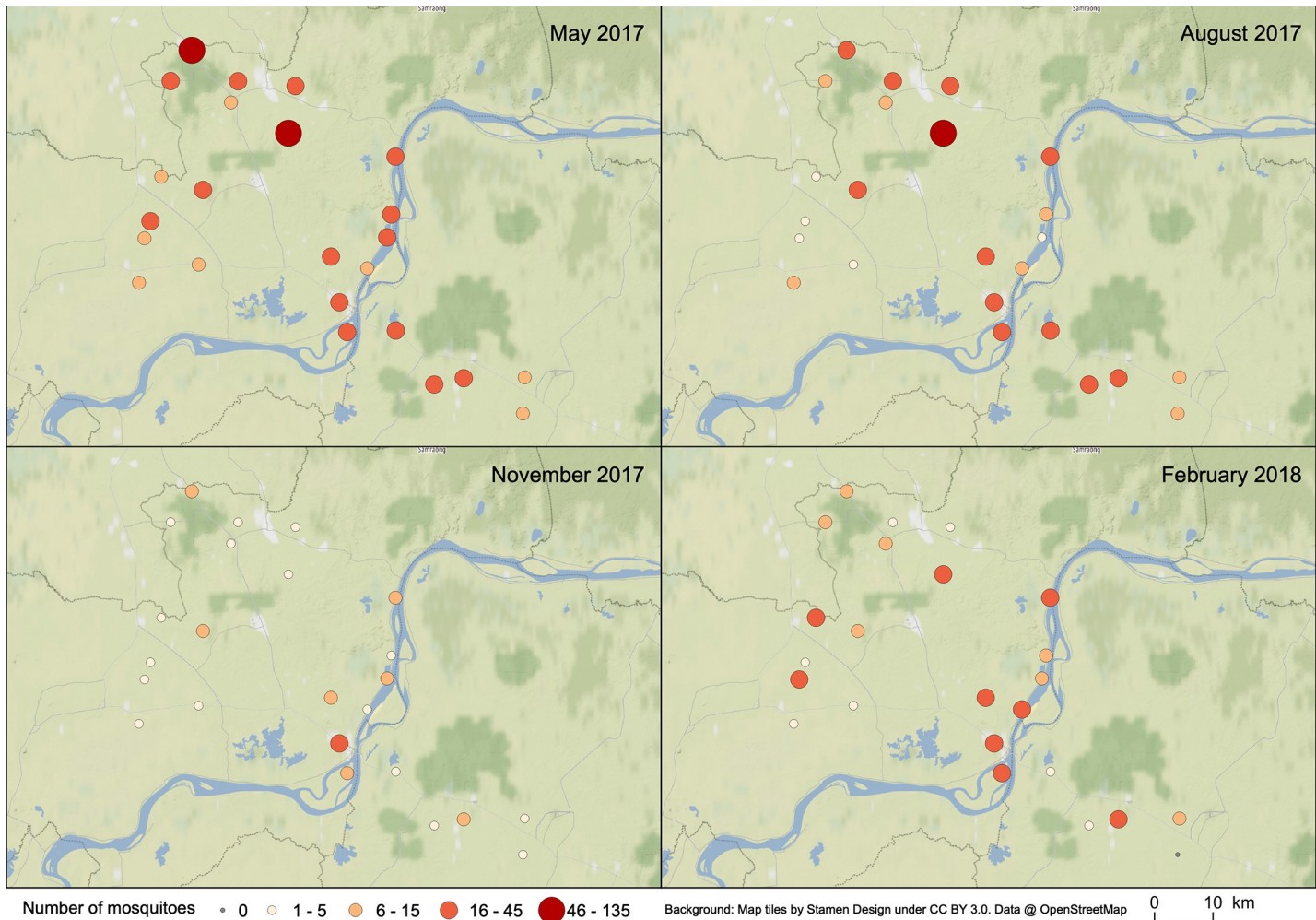

**Fig 3. Spatial and temporal distribution of the relative abundance of dengue virus vectors in the 24 schools in Kampong Cham and Tboung Khmum provinces, Cambodia.**

density in the majority of schools (with only 3 schools recording less than 100 mosquitoes with a minimum average of 67 JEV vectors per 4 days), especially important in three schools mainly in Angkor (relative density of 1,021 JEV vectors per 4 days), Koh Penh (796) and Punley (648) schools (Fig 4C).

## Discussion

### Diversity of mosquito species

There was a large overall diversity of mosquito species caught across all schools and even reached 61 different mosquito species in one school. Based on the various specimens deposited in institutions for collections (IRD and Smithsonian), we estimated that at least 243 mosquito species are present in Cambodia. This estimate of the mosquito fauna biodiversity is important, but it may be underestimated as neighboring countries such as Thailand officially recorded 384 species [18]. Other neighboring countries such as Lao PDR and Vietnam have described recently 170 and 191 taxa, respectively [19; 20; 21], but as in Cambodia they are probably underestimating the mosquito diversity. By collecting 61 species in 24 schools during

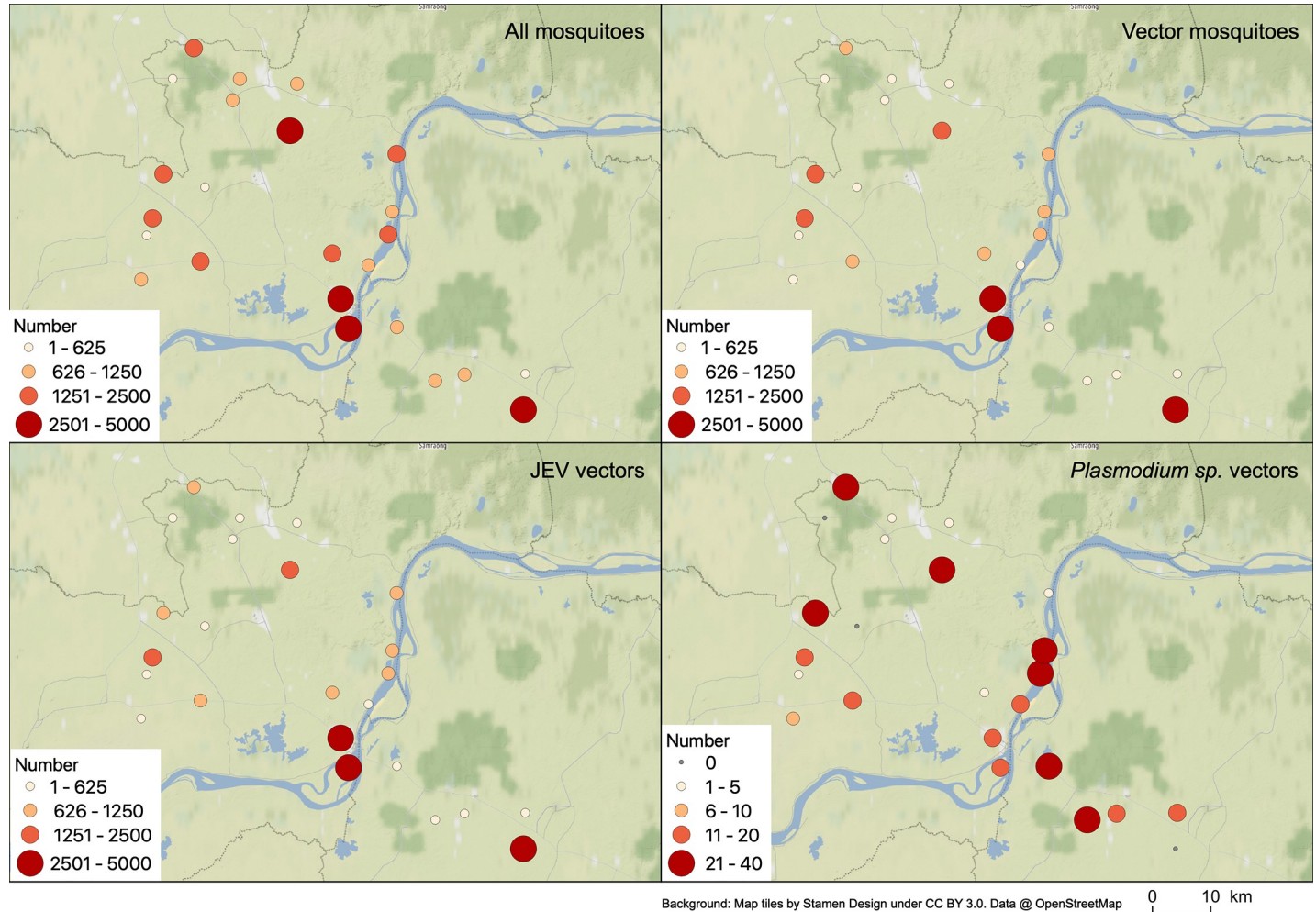

**Fig 4.** Spatial distribution of the relative abundance of (A) all mosquitoes, (B) vector mosquitoes, (C) JEV vectors and (D) *Plasmodium sp*. vectors in the 24 schools in Kampong Cham and Tboung Khmum provinces, Cambodia.

our study, representing 23% of the country's species, we probably underestimate the number of mosquito species present in schools in Cambodia. Insofar as the majority of the species collected have been previously incriminated as vectors of pathogens of public health importance, a global inventory of the biodiversity of mosquitoes in schools should be undertaken in Cambodia and control strategies developed.

Little is known on the epidemiological importance of schools in arbovirus and other pathogen transmission. There are only few studies worldwide, describing the mosquito biodiversity in these specific places, thus making it difficult to compare with our results (i.e. number and index). For example, a study implemented in Thailand in 2012 in classrooms only showed the numbers of *Ae. aegypti* collected without describing the other potential vector species present. The authors described the presence of 66 adult *Ae. aegypti* collected in 10 schools from 50 classrooms, after the use of vacuum aspirators for 15 minutes per room [22]. The total number of mosquitoes (278) was very low compared to our methods, likely due to the methodology used, even if the proportion of *Ae. aegypti* (24%) is higher [22]. In Colombia, during a surveillance carried out in 34 schools, adult mosquitoes were collected twice a year during 10 minutes

with electric aspirators in 191 and 188 classrooms in 2 municipalities, and 9 mosquito species were formally described including *Aedes*, *Anopheles*, *Culex* and other genera. [23]. However, due to the geography, climate and other factors, it seems incoherent to compare our results to those of Colombia, where 324 mosquito species from 28 genera were recently described [24].

## Diversity and relative abundance of mosquito potential vector species

Children spend their time at school during the day and consequently these specific places are most likely to be considered as hot spots for transmission of vector borne-diseases. In our study, almost nine out of ten mosquitoes could be considered as a potential vector and confirms the interest of carrying out surveillance or even vector control in and around schools in highly endemic diseases areas. It should also be noted that 17 species present in the 24 schools can be considered as a potential vectors.

The presence of *Ae. aegypti* and *Ae. albopictus* in the 24 schools and during the 4 collection sessions implemented throughout the year illustrates the role that these species could play in the maintenance and transmission of viruses such as dengue in schools in general. In Thailand, *Ae. aegypti* was also collected in all rural, semi-rural and semi-urban schools [22]. Moreover, in the studies carried out in Thailand and Colombia, in both cases the majority of *Ae. aegypti* females were collected in the classrooms, as in our study, and many mosquitoes were positive for the dengue by PCR detection confirming their possible role [23; 22]. Moreover, these mosquitoes are known for their diurnal biting activity, increasing the chance to play a key role in schools for the transmission or maintenance of dengue fever in endemic areas.

A large majority of JEV vectors was observed with the notable presence of *Cx. vishnui*, *Cx. quinquefasciatus*, *Cx. gelidus* and also of considered secondary vectors such as *Cx. fuscocephala*, *Cx. tritaeniorhynchus* and *Cx. bitaeniorhynchus*. The proportion of these vectors (70%) in schools is comparable to the relative density found in a study of mosquito dynamics in rural and peri-urban areas in Cambodia (Boyer Pers. Com.). The presence of large numbers of these vectors in classrooms could also explain the high endemicity of Japanese encephalitis in Cambodia [8]. The large number of JEV vector species caught in classrooms suggests an anthropophagic behavior, and yet the main vectors of JEV are often considered to be highly zoophilic, accidentally biting humans [25, 26; 27; 28]. With their presence within the classrooms, the consideration of these species as opportunistic [28] seems more appropriate in Cambodia. However, contrary to *Ae. aegypti* and *Ae. albopictus*, *Culex* spp. biting time is described as being nocturnal, which may minimize the importance of these potential mosquitoes in the transmission.

Also considered as night biting mosquitoes, 6 secondary vector species of *Plasmodium* spp. were described in 20 of the 24 schools, namely *An. barbirostris*, *An. barbumbrosus*, *An. campestris*, *An. philippinensis*, *An. sinensis* and *An. vagus*. Considering the total number of these mosquitoes (325 mosquitoes representing only 0.91%), their role as only secondary vectors, the fact that malaria transmission occurs in the forest in Cambodia [29], the risk of malaria transmission is considered very low. Nevertheless, it is important to describe their presence in schools for any future evolution or dynamic of *Plasmodium* transmission.

## Conclusion

The important biodiversity of mosquitoes discovered in schools in Cambodia is relevant and can represent a research axis, especially on the association between the different surrounding ecotypes (forest, culture, rice fields and urbanity) and the presence of specific mosquito species or vectors. Spatio-temporal analysis to study landscapes and weather effects should be considered to understand the distribution and abundance of different species in schools.

Such diversity was unexpected and inevitably leads to a diversity of potential vectors. The diversity of vectors, their dynamics, relative abundance, and distribution must be analyzed with mosquito behavioral studies to estimate the risk of transmission by evaluating the vector–schoolchildren/teachers contact. To be exhaustive, a screening of pathogens carried by mosquitoes in schools should be implemented. The importance of JEV and DENV vectors in schools is in accordance with the main diseases circulating in Cambodia and affecting children. These findings are of particular importance and will help to recommend appropriate vector control strategies and program activities regarding specific locations such as schools and their environmental settings. Moreover, the description of the important presence of vectors in schools is an important step prior to vector control and for testing the implementation of new control methods adapted to schools.

## Supporting information

**S1 Data. List of mosquito species caught in the different schools.**
(XLSX)

**S1 File. Results of Generalized Linear Mixed Model (GLMM) using Template Model Builder (TMB) using Family Poisson with 'school' as a random effect reflecting the random choice of the schools on all mosquitoes, all vector mosquitoes, Japanese encephalitis virus vectors, Dengue virus vectors and Plasmodium spp. Vectors.**
(DOCX)

## Acknowledgments

The authors really want to thank the technicians that did the sampling in the field, Suor Kimhuor, Chhum Moeun, Chhuoy Kalyan and the different field authorities including all the Directors and Teachers of the 24 primary schools. The authors also would like to thank Sylvaine Jego for statistical advices, Vincent Herbreteau for the help for mapping, and Richard Paul for correcting the manuscript.

## Author Contributions

**Conceptualization:** Sebastien Boyer, Didier Fontenille.

**Data curation:** Sebastien Boyer, Sony Yean.

**Formal analysis:** Sebastien Boyer.

**Funding acquisition:** Sebastien Boyer, Didier Fontenille.

**Investigation:** Sebastien Boyer, Sony Yean.

**Methodology:** Sebastien Boyer, Sony Yean, Didier Fontenille.

**Project administration:** Sebastien Boyer, Didier Fontenille.

**Resources:** Sebastien Boyer, Didier Fontenille.

**Software:** Sebastien Boyer.

**Supervision:** Sebastien Boyer.

**Validation:** Sebastien Boyer.

**Visualization:** Sebastien Boyer.

**Writing – original draft:** Sebastien Boyer, Sebastien Marcombe, Sony Yean, Didier Fontenille.

**Writing – review & editing:** Sebastien Boyer, Sebastien Marcombe, Didier Fontenille.

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
