## [Decision Letter · Decision Letter 0]

18 Sep 2019

PONE-D-19-20103

High diversity of mosquito vectors in Cambodian primary schools and consequences for arboviruses transmission

PLOS ONE

Dear Dr. Boyer,

Thank you for submitting your manuscript to PLOS ONE. After careful consideration, we feel that it has merit but does not fully meet PLOS ONE’s publication criteria as it currently stands. Therefore, we invite you to submit a revised version of the manuscript that addresses the points raised during the review process. Note that both reviewers have concern about the English and therefore I request that you consult either a native speaker or a proofreading company before re-submission.

We would appreciate receiving your revised manuscript by Nov 02 2019 11:59PM. To enhance the reproducibility of your results, we recommend that if applicable you deposit your laboratory protocols in protocols.io, where a protocol can be assigned its own identifier (DOI) such that it can be cited independently in the future. For instructions see: http://journals.plos.org/plosone/s/submission-guidelines#loc-laboratory-protocols

We look forward to receiving your revised manuscript.

Kind regards,

Olle Terenius

Academic Editor

PLOS ONE

2.In your Methods section, please provide additional information regarding the permits you obtained for the work. Please ensure you have included the full name of the authority that approved the field site access and, if no permits were required, a brief statement explaining why

a) You may seek permission from the original copyright holder of Figure 1 to publish the content specifically under the CC BY 4.0 license.  

4. Please include your tables as part of your main manuscript and remove the individual files. Please note that supplementary tables (should remain/ be uploaded) as separate "supporting information" files.

Reviewers' comments:

Reviewer's Responses to Questions

**Comments to the Author**

1. Is the manuscript technically sound, and do the data support the conclusions?

Reviewer #1: Partly

Reviewer #2: No

2. Has the statistical analysis been performed appropriately and rigorously? 

Reviewer #1: Yes

Reviewer #2: No

3. Have the authors made all data underlying the findings in their manuscript fully available?

Reviewer #1: No

Reviewer #2: Yes

4. Is the manuscript presented in an intelligible fashion and written in standard English?

Reviewer #1: Yes

Reviewer #2: No

5. Review Comments to the Author

Reviewer #1: REVIEW COMMENTS MANUSCRIPT NUMBER PONE-D-19-20103; High Diversity of Mosquito vectors in Cambodia primary schools and consequences for arbovirus transmission

Overall general comments

• The research sets out to evaluate the vector abundance and diversity in primary schools in part of Cambodia in an attempt to explain the apparent burden of dengue fever, severe dengue (DEN), Japanese encephalitis (JE), other arboviruses and parasites like malaria among children, 15 years and below attending selected primary schools through vector surveys.

• In the introduction the study is well justified and rationalized. The data collection tools though not exhaustive (lacking in the survey of school environment for breeding sites) were appropriate and the data generated was adequate to address the set objective.

• On the analysis of data, emphasis has been placed on expressing the diversity indices. The initial objective of explaining the burden of dengue and other arboviruses using the vector data in the diverse schools was clouded by the overemphasis on diversity indices. Additionally, interpretation of the diversity indices to explain the specific disease burden was not clearly brought (how will these indices inform decision towards reducing disease burden?). However, abundance of specific species to explain transmission of the target diseases, DEN, JE and malaria has been articulated

• Key data sets mentioned in the manuscript (table 2) could not be found in the submitted manuscript hence the data referred to could not be verified to confirm the claims.

• There is mention of an attempt to relate school characteristics (supposedly, the physical structures of the schools) to vector data but the table (table 1) referred to, which describes the characteristics could not be found in the manuscript. In addition, the data analysis presented in the results made no reference to the said school characteristics and weather there was any correlation to the vector data abundance or diversity.

Based on the above important major observation, the authors need to work further on the manuscript to improve on the listed issues and gaps before it can be considered further.

Specific comments Major

Abstract

• The abstract is too thin on specific information that would be needed to get the real picture of what was found in terms of the specific target diseases. The information has been overgeneralized.

• The conclusion missed the key point of how the data explains burden of the target diseases and what can be done about it. There was focus on biodiversity as a conclusion.

Introduction

• The introduction has relevant information that pertains to the subject of investigation but its presentation is marred by numerous grammatical mistakes. It is suggests that if the authors are not usual English writers, they may need to identify someone to edit the manuscript for grammar.

• Apart from Dengue, there is little information on the other arbovirus diseases like CHIK and JE and how they impacts on the population under 15. How is the incidence of DEN, CHIK and JE in the general population that suggests that children are disproportionately affected to rationalize focus on this age group.

Methods

• How were the school characteristics classified and how were these used to define vector distribution and /or abundance?

• How were the mosquitoes handled after trapping before identification? Were they identified in the field or in the lab? How were they transported? The methodology is very scanty with information.

• Where were the CDC traps located within the school?

Results

• As you discuss your data, explain how diversity indices measured (simpson and Shannon) influence the risk of dengue, JE or malaria transmission which seemed to have informed the selection of schools as implementation sites, arising from the burden of DEN and JE among children 15ys and below.

• The tables detailing the data described.in the results is not accessible.

Reviewer #2: The manuscript entitled “High diversity of mosquito vectors in Cambodian primary schools and consequences for arboviruses transmission” determined the mosquito diversity and abundance in 24 schools from two different provinces in Cambodia during the dry and rainy seasons in 2017-2018. The authors evaluated the mosquito diversity using light traps and BG traps Shannon and Simpson indexes. In the abstract, 61 species were described in schools, including Aedes, Anopheles and Culex genus. The manuscript proposal is interesting and promising to Cambodia authorities, especially because the authors mention that there are few data available on mosquito diversity in the country. However, the present manuscript needs to be thoroughly rewritten and more details should be given regarding experimental design, data analysis, results (e.g., Tables 1 and 2 are not present in the manuscript version that was sent to me), among others.

The authors should be more didactic in the analysis section: what were the independent variables? The authors mention that they analyze the data using a “The influences of parameters on Shannon and Simpson indexes were first determined by a general linear model with a stepwise method (lines 99-100)”. What was the model? Was it a GLM? What probability distribution was used? What are the “parameters” that authors mention? Why did they use two models to analyze the same data? How the authors managed to overcome pseudoreplication problems that are common with this kind of study? This section is confusing and is not clear. The authors should rethink their analysis strategy and probably update their results. The results are also very confusing as the authors use absolute abundance, relative abundance, the diversity indexes for the schools and seasons interchangeably and do not give depth to their analysis. They should focus on their results separating by sections. The results could be more explored, and they are not suitable for PloS One such as they are presented. I suggest the authors to explore a spatial approach in their data so that they have more interesting material. The discussion and conclusion sections are very interesting but should be updated after a new batch of analysis are done. Also, there are lots English mistakes and typos that need revision.

6. PLOS authors have the option to publish the peer review history of their article (what does this mean?). If published, this will include your full peer review and any attached files.

Reviewer #1: No

Reviewer #2: No

---

## [Author Response · Author response to Decision Letter 0]

3 Dec 2019

PLOS ONE. ONE-D-19-20103 : High diversity of mosquito vectors in Cambodian primary schools and consequences for arbovirus transmission

ANSWERS TO EDITORIAL COMMENTS AND REVIEWERS

ANSWER TO EDITORIAL COMMENTS

1. Journal requirements

We changed the formatting of the main body and of the authors’ affiliation.

2. In your Methods section, please provide additional information regarding the permits you obtained for the work. Please ensure you have included the full name of the authority that approved the field site access and, if no permits were required, a brief statement explaining why

LINES 79-84:

You can now find the following sentences: “The National Ethics Committee for Health Research of the Ministry of Health of the Kingdom of Cambodia approved the field site access and the permit N° 113NECHR was obtained on 02 May 2017. The study was also fully supported by the National Center for Parasitology, Entomology and Malaria Control, Ministry of Health (Official mail N° 348/17 dated from 28 April 2017), and by the Ministry of Education, Youth and Sport (Official mail N° 2592 dated from 27 April 2017).”

 a) You may seek permission from the original copyright holder of Figure 1 to publish the content specifically under the CC BY 4.0 license. 

The figure 1 is a homemade map with no copyright. This map was never copyrighted until now and was never publish. We did not use any satellite image or copyrighted background. We did ourselves this vectorial map by using a public administrative contour map. And we recorded school location by using a GPS.

4. Please include your tables as part of your main manuscript and remove the individual files. Please note that supplementary tables (should remain/ be uploaded) as separate "supporting information" files.

Tables 1 & 2 were incorporated in the main manuscript in the Results part.

We also add a Table 3 directly in the manuscript

ANSWER TO REVIEWERS’ COMMENTS

For the overall general comments, Reviewer 1 emphases our introduction and Reviewer 2 the discussion and conclusion, recommending to focus on Public Health areas. It seems that the biodiversity indices that we wanted to introduce through this article, bring too much perturbation to this article. We removed these common indices of biodiversity, but still talking about the unexpected biodiversity of mosquito species we found inside class rooms. Overall we add 3 tables in the text, we removed 4 figures and replaced them by 3 other.

Reviewer #1: 

Overall general comments

• The research sets out to evaluate the vector abundance and diversity in primary schools in part of Cambodia in an attempt to explain the apparent burden of dengue fever, severe dengue (DEN), Japanese encephalitis (JE), other arboviruses and parasites like malaria among children, 15 years and below attending selected primary schools through vector surveys.

• In the introduction the study is well justified and rationalized. The data collection tools though not exhaustive (lacking in the survey of school environment for breeding sites) were appropriate and the data generated was adequate to address the set objective.

• On the analysis of data, emphasis has been placed on expressing the diversity indices. The initial objective of explaining the burden of dengue and other arboviruses using the vector data in the diverse schools was clouded by the overemphasis on diversity indices. Additionally, interpretation of the diversity indices to explain the specific disease burden was not clearly brought (how will these indices inform decision towards reducing disease burden?). 

We deleted the results parts and the figures and maps related to the diversity indices that are not related to the different diseases and induced more confusion. But, we keep a result and discussion part on the diversity, even if we decrease the importance of this part, in order to illustrate also the huge diversity of mosquito species in the schools.

I am totally agree that there is no relationship between the diversity of all mosquito species and the different diseases. 

However, abundance of specific species to explain transmission of the target diseases, DEN, JE and malaria has been articulated.

• Key data sets mentioned in the manuscript (table 2) could not be found in the submitted manuscript hence the data referred to could not be verified to confirm the claims.

We are truly sorry with this major problem. AS usual, we add the tables as Excel files. But, in PLOS ONE, they ask to add the tables inside the text. It seems that the editors didn’t check it and send you the file without the tables. We feel terribly sorry for the reviewers, and also a little disappointed against the editors work.

• There is mention of an attempt to relate school characteristics (supposedly, the physical structures of the schools) to vector data but the table (table 1) referred to, which describes the characteristics could not be found in the manuscript. In addition, the data analysis presented in the results made no reference to the said school characteristics and weather there was any correlation to the vector data abundance or diversity.

These analysis were not made yet. The characterization of the landscape is very difficult and takes time. Regarding the objective characteristics obtained in the schools, such as the characteristics of the school, we wrote lines 108-117 “The association of the different parameters characterizing the schools and the period of collection, i.e. month, school, urbanization, district, province, presence of pagoda, school area, the number of classroom and children in the school and the relative densities of mosquito populations (i.e. all mosquitoes, all vector mosquitoes, Japanese encephalitis virus vectors, Dengue virus vectors, malaria plasmodium vectors) in the considered schools, were tested by a stepwise algorithm model, with backward elimination of non-significant parameters until a final minimum adequate model containing only significantly associated variables, in order to determine the main significant factors with the confirmation of residuals following the normality. A deeper characterization of month and school effects and their comparisons was carried out by using a Kruskal-Wallis test”.

Based on the above important major observation, the authors need to work further on the manuscript to improve on the listed issues and gaps before it can be considered further.

As you recommended, we reoriented the manuscript in regards of the observations you highlighted. Particularly, we focused more on the vector species, and we developed the conclusion on the possible ways of research.

Specific comments Major

Abstract

• The abstract is too thin on specific information that would be needed to get the real picture of what was found in terms of the specific target diseases. The information has been overgeneralized.

In the abstract we add few sentences. A first sentence related to the main picture: “The research sets out to evaluate the vector relative abundance and diversity in primary schools in part of Cambodia in an attempt to explain the apparent burden of dengue fever, severe dengue (DEN), Japanese encephalitis (JE), other arboviruses and parasites like malaria among children, 15 years and below attending selected primary schools through vector surveys.”.

And we also add one sentence on the main results in terms of Public Health problematic: “Our results showed that schools in Cambodia represent a potential risk of vector born disease transmission and highlight the importance to implement vector control in schools in Cambodia to decrease the risk of transmission.”

• The conclusion missed the key point of how the data explains burden of the target diseases and what can be done about it. There was focus on biodiversity as a conclusion. 

In the conclusion as at the end of the abstracts, we focused now mainly on the importance of implementing new methods for vector controls.

Introduction

• The introduction has relevant information that pertains to the subject of investigation but its presentation is marred by numerous grammatical mistakes. It is suggests that if the authors are not usual English writers, they may need to identify someone to edit the manuscript for grammar.

The article was read again by a French native speaker that lived in U.S., and then, the article was read by an entomologist, Dr. Richard Paul, who is a English native speaker. 

• Apart from Dengue, there is little information on the other arbovirus diseases like CHIK and JE and how they impacts on the population under 15. How is the incidence of DEN, CHIK and JE in the general population that suggests that children are disproportionately affected to rationalize focus on this age group. 

We choose to let these information in the Discussion part. But we had a sentence regarding the incidence of Dengue in children in Cambodia. About Japanese encephalitis, there is no data in Cambodia on the incidence of JEV in the population, incidence of JEV in children, neither a repartition of percentage children/total population. The only recent study is on the causes of acute meningoencephalitis in children population (with 24.4% of JEV). But there is no global data to add in the introduction. 

Methods

• How were the school characteristics classified and how were these used to define vector distribution and /or abundance?

The school characteristics are group in Table 1. You did not have this Table 1 in the previous version of the manuscript. I am sorry for that. Now, you can have access to these data. They were tested but none of them was influencing the data. There is another huge work to do for each species related to each factors, including temperature and spatial effect, but this is another huge analysis work that we need to begin.

• How were the mosquitoes handled after trapping before identification? Were they identified in the field or in the lab? How were they transported? The methodology is very scanty with information.

LINES 95-105, we specified and detailed more our field protocol:

“Traps were set up during 24 hours inside classrooms with 1 trap per classroom. Following the traditional Cambodian construction, all classes of all schools have openings in the tops of each wall to let the air circulates and reduce the high temperatures occurring during the daytime, allowing the free circulation of mosquitoes. Depending on the size of the schools and the number of classrooms, three (or less) BG traps and three (or less) CDC light traps were used with a limit of a total of six traps per school (Table 1). After each collection, mosquitoes were conserved in an electric icebox at +4°C and first identified in the field the same day. Then, samples were brought back to the laboratory, still at +4°C, in Phnom Penh for a second identification. In the field and in the laboratory, mosquitoes were identified under stereomicroscope and microscope using morphological mosquito identification keys from Southeast Asia countries (Rattanarithikul et al. 2005; Stojanovich, and Scott 1966).”

• Where were the CDC traps located within the school?

As writing in the previous sentence: “inside the classrooms”. But we also precised why : “raps were set up during 24 hours inside classrooms with 1 trap per classroom. Following the traditional Cambodian construction, all classes of all schools have openings in the tops of each wall to let the air circulates and reduce the high temperatures occurring during the daytime, allowing the free circulation of mosquitoes.” (LINES 95-99)

Results

• As you discuss your data, explain how diversity indices measured (simpson and Shannon) influence the risk of dengue, JE or malaria transmission which seemed to have informed the selection of schools as implementation sites, arising from the burden of DEN and JE among children 15ys and below.

• The tables detailing the data described.in the results is not accessible.

The diversity (as far as I know) did not influence the risks of dengue, Japanese encephalitis and so on. It doesn’t mean it didn’t, but I don’t think so, and it was never prove. As I wrote before, regarding the comments of the reviewers, we removed the parts on the biodiversity indices to focus on the vector species.

And again, I really want to apologize for the absence of the table within the manuscript.

Reviewer #2: 

The manuscript entitled “High diversity of mosquito vectors in Cambodian primary schools and consequences for arboviruses transmission” determined the mosquito diversity and abundance in 24 schools from two different provinces in Cambodia during the dry and rainy seasons in 2017-2018. The authors evaluated the mosquito diversity using light traps and BG traps Shannon and Simpson indexes. In the abstract, 61 species were described in schools, including Aedes, Anopheles and Culex genus. The manuscript proposal is interesting and promising to Cambodia authorities, especially because the authors mention that there are few data available on mosquito diversity in the country. However, the present manuscript needs to be thoroughly rewritten and more details should be given regarding experimental design, data analysis, results (e.g., Tables 1 and 2 are not present in the manuscript version that was sent to me), among others.

Following the recommendations, Tables 1, 2 & 3 are now included within the body text of the article. As for the reviewer #1, I really want to apologize for the absence of the 2 previous tables. Generally, the tables are add in Excel files, but PLOS asked to put into the text. Deeply sorry for that inconvenient.

Following recommendations, some details were added in Material and Methods sections for experimental design, data analysis and also in the results part, as detailed before and below. 

The authors should be more didactic in the analysis section: what were the independent variables? The authors mention that they analyze the data using a “The influences of parameters on Shannon and Simpson indexes were first determined by a general linear model with a stepwise method (lines 99-100)”. What was the model? Was it a GLM? 

Yes. It was written ‘general linear model’ meaning GLM.

What probability distribution was used? We used the What are the “parameters” that authors mention? Why did they use two models to analyze the same data? How the authors managed to overcome pseudoreplication problems that are common with this kind of study? This section is confusing and is not clear. The authors should rethink their analysis strategy and probably update their results. 

The section statistical analysis was totally rewritten. You can no find lines 119-126:

“The influences of the different parameters characterizing the schools and the period of capture, i.e. month, school, urbanization, district, province, presence of pagoda, school area, the number of classroom and children in the school, the population in the considered village, were tested by first a general linear model (glm) with the confirmation of residuals following the normality, (2) then by a stepwise algorithm model to determine the main significant factors. None were significant except the month and school effects. A deeper characterization of month and school effects and their comparisons were realized by using a Kruskal-Wallis test.”

A Table 3 was added in the Result section to update the results, and discussed them.

The results are also very confusing as the authors use absolute abundance, relative abundance, the diversity indexes for the schools and seasons interchangeably and do not give depth to their analysis. They should focus on their results separating by sections. The results could be more explored, and they are not suitable for PloS One such as they are presented. I suggest the authors to explore a spatial approach in their data so that they have more interesting material. The discussion and conclusion sections are very interesting but should be updated after a new batch of analysis are done. Also, there are lots English mistakes and typos that need revision.

---

## [Decision Letter · Decision Letter 1]

17 Mar 2020

PONE-D-19-20103R1

High diversity of mosquito vectors in Cambodian primary schools and consequences for arboviruses transmission

PLOS ONE

Dear Dr. Boyer,

Thank you for submitting your manuscript to PLOS ONE. After careful consideration, we feel that it has merit but does not fully meet PLOS ONE’s publication criteria as it currently stands. Therefore, we invite you to submit a revised version of the manuscript that addresses the points raised during the review process.

Please respond to the reviewers' comments. Additional experiments are not needed, but adjustment of the modelling can be warranted.

We would appreciate receiving your revised manuscript by May 01 2020 11:59PM. To enhance the reproducibility of your results, we recommend that if applicable you deposit your laboratory protocols in protocols.io, where a protocol can be assigned its own identifier (DOI) such that it can be cited independently in the future. For instructions see: http://journals.plos.org/plosone/s/submission-guidelines#loc-laboratory-protocols

We look forward to receiving your revised manuscript.

Kind regards,

Olle Terenius

Academic Editor

PLOS ONE

Reviewers' comments:

Reviewer's Responses to Questions

**Comments to the Author**

1. If the authors have adequately addressed your comments raised in a previous round of review and you feel that this manuscript is now acceptable for publication, you may indicate that here to bypass the “Comments to the Author” section, enter your conflict of interest statement in the “Confidential to Editor” section, and submit your "Accept" recommendation.

Reviewer #3: (No Response)

Reviewer #4: (No Response)

2. Is the manuscript technically sound, and do the data support the conclusions?

Reviewer #3: Partly

Reviewer #4: Partly

3. Has the statistical analysis been performed appropriately and rigorously? 

Reviewer #3: I Don't Know

Reviewer #4: No

4. Have the authors made all data underlying the findings in their manuscript fully available?

Reviewer #3: (No Response)

Reviewer #4: Yes

5. Is the manuscript presented in an intelligible fashion and written in standard English?

Reviewer #3: (No Response)

Reviewer #4: Yes

6. Review Comments to the Author

Reviewer #3: Authors have collected mosquitoes in school and have observed high diversity. Some of mosquitoes collected are known to be vectors. However, authors don't report any evidence of presence of parasite or virus in mosquitoes collected but have largely discussed the transmission risk in schools. Since it is possible (not too difficult) to detect parasites (e.g. Plasmodium spp) and virus (e.g. DENV and other) in mosquitoes, I strongly suggest to authors to pool mosquitoes collected per species per school in order to search parasites and virus in these mosquitoes. Then, authors would be more consistent with there conclusion.

Reviewer #4: Overall, this manuscript is very important in providing the mosquito diversity in the country especially because authors mention that there are few data available on mosquito diversity in the country. In understanding that it might be referenced a lot in the future once published - it is important to make sure the manuscript is well written, the data analysis is appropriately and well performed, and the recommendations for vector control options are very clearer.

Anyway, authors have attempted to the large extent to address reviewers' comments. However, more work still need to be done, especially on the data analysis and recommendations of what needs and can be done.

Given that there are so many random effects – I would have expected the analysis to be done using a generalized linear mixed model (GLMM) which takes into account random effects in addition to the usual fixed effects rather than using GLM. The statistician should check the analysis and the numbers presented. It might be the case that the characterization of the landscape is very difficult and takes time but incorporation of seasonality shouldn't take time.

This manuscript could have been strong if the study also looked at the breeding sites in school surroundings and its correlations with mosquitoes caught in the classroom.

Authors should provide a clear recommendations of the vector control options and their justification. The options should be discussed in the discussion section stating why they are expected to work, how they will be implemented? etc? Especially in such a setting (i.e., classroom - school environment)

7. PLOS authors have the option to publish the peer review history of their article (what does this mean?). If published, this will include your full peer review and any attached files.

Reviewer #3: No

Reviewer #4: No

---

## [Author Response · Author response to Decision Letter 1]

20 Apr 2020

PLOS ONE. ONE-D-19-20103 R1: High diversity of mosquito vectors in Cambodian primary schools and consequences for arbovirus transmission

ANSWERS TO REVIEWERS

Reviewer #3

Authors have collected mosquitoes in school and have observed high diversity. Some of mosquitoes collected are known to be vectors. However, authors don't report any evidence of presence of parasite or virus in mosquitoes collected but have largely discussed the transmission risk in schools. Since it is possible (not too difficult) to detect parasites (e.g. Plasmodium spp) and virus (e.g. DENV and other) in mosquitoes, I strongly suggest to authors to pool mosquitoes collected per species per school in order to search parasites and virus in these mosquitoes. Then, authors would be more consistent with their conclusion.

We agree with the reviewer comments about virus/parasites detection in the mosquitoes collected. This is mainly a financial issue. We hope to have more projects in the future to be able to detect what is really circulating. Even if the entomological surveillance is difficult, with an estimation of less than 1 positive mosquito on 1,000 mosquitoes estimated, even during outbreak (few examples exist: Rift Valley fever in Madagascar, Zika in French Guyana…).

One sentence was added in the conclusion line 265: “To be exhaustive, a screening of pathogens carried by mosquitoes in schools should be implemented”.

Reviewer #4

Overall, this manuscript is very important in providing the mosquito diversity in the country especially because authors mention that there are few data available on mosquito diversity in the country. In understanding that it might be referenced a lot in the future once published - it is important to make sure the manuscript is well written, the data analysis is appropriately and well performed, and the recommendations for vector control options are very clearer.

Anyway, authors have attempted to the large extent to address reviewers' comments. However, more work still need to be done, especially on the data analysis and recommendations of what needs and can be done.

Given that there are so many random effects – I would have expected the analysis to be done using a generalized linear mixed model (GLMM) which takes into account random effects in addition to the usual fixed effects rather than using GLM. The statistician should check the analysis and the numbers presented. It might be the case that the characterization of the landscape is very difficult and takes time but incorporation of seasonality shouldn't take time.

Thank you for this remark. The reviewer was right: We indeed go back to the data by using glmm (function glmmTMB) in R, and observe significant effects, even if slight when looking at the data.

In fact, we have randomly selected our schools at the beginning, without taking into account the representativeness of all the schools in the region studied according to different parameters (number of students, pop dependent on this school, primary / college, urban / rural,% patients with the virus in the study area (or 5 km away, etc.). This is why we considered that the school is random. In fact, by putting the school in random effect, the GLMM calculated a regression for each school and the coefficients estimated for the other variables without the school bias. 

At the end of the analysis, we repeat a shapiro.test on the residues, and we found a better result with residuals closer to normal. We also test the overdistention (surdispersion) of data and corrected it.

To summary, we finally did a glm analysis by taking schools as random effect.

The other random effect could have be the month of capture because, for organizational reasons, we did samplings every 3 months. So we also tested the month as a random factor. But there was no difference in the quality of the model (almost the same AIC) and we chose to include the month in the variables to be able to study its effect. 

For the other point, as the observational study is on only one year, we are not able to study the seasonality as there is no replicates in time. We hope to be able to work on a several years’ observation database.

We precised in the text (Material and Method part, Statistical Analysis section) which analysis we did. We changed all this part. Regarding the results, we added the statistical effect in the text. Moreover we add a Supplementary file for the results of the analysis.

This manuscript could have been strong if the study also looked at the breeding sites in school surroundings and its correlations with mosquitoes caught in the classroom.

We also agree with that. We observed few breeding sites (in toilets schools, and few plastic garbage) for looking more specifically at Dengue vectors, but we did not do a complete characterization of all breeding sites in 24 schools (logistic and financial reasons). Also, this would have been another full study. 

Authors should provide a clear recommendations of the vector control options and their justification. The options should be discussed in the discussion section stating why they are expected to work, how they will be implemented? etc? Especially in such a setting (i.e., classroom - school environment)

We remained global regarding recommendations for vector control because we did not study or characterize the insecticide resistance of the most dominant species, furthermore, we did not study their biology including the breeding sites as mentioned earlier.

---

## [Decision Letter · Decision Letter 2]

12 May 2020

High diversity of mosquito vectors in Cambodian primary schools and consequences for arboviruses transmission

PONE-D-19-20103R2

Dear Dr. Boyer,

We are pleased to inform you that your manuscript has been judged scientifically suitable for publication and will be formally accepted for publication once it complies with all outstanding technical requirements.

With kind regards,

Olle Terenius

Academic Editor

PLOS ONE

Additional Editor Comments (optional):

Reviewers' comments:

Reviewer's Responses to Questions

**Comments to the Author**

1. If the authors have adequately addressed your comments raised in a previous round of review and you feel that this manuscript is now acceptable for publication, you may indicate that here to bypass the “Comments to the Author” section, enter your conflict of interest statement in the “Confidential to Editor” section, and submit your "Accept" recommendation.

Reviewer #3: All comments have been addressed

Reviewer #4: All comments have been addressed

2. Is the manuscript technically sound, and do the data support the conclusions?

Reviewer #3: Yes

Reviewer #4: Yes

3. Has the statistical analysis been performed appropriately and rigorously? 

Reviewer #3: I Don't Know

Reviewer #4: I Don't Know

4. Have the authors made all data underlying the findings in their manuscript fully available?

Reviewer #3: Yes

Reviewer #4: Yes

5. Is the manuscript presented in an intelligible fashion and written in standard English?

Reviewer #3: Yes

Reviewer #4: Yes

6. Review Comments to the Author

Reviewer #3: The main comment of my review was relative to the detection of parasites in mosquitoes collected. Authors declared that they are agree with my comment and have added a sentence in the conclusion.

Reviewer #4: (No Response)

7. PLOS authors have the option to publish the peer review history of their article (what does this mean?). If published, this will include your full peer review and any attached files.

Reviewer #3: No

Reviewer #4: No

---

## [Editor Report · Acceptance letter]

20 May 2020

PONE-D-19-20103R2 

High diversity of mosquito vectors in Cambodian primary schools and consequences for arbovirus transmission 

Dear Dr. Boyer:

I am pleased to inform you that your manuscript has been deemed suitable for publication in PLOS ONE. Congratulations! Your manuscript is now with our production department. 

With kind regards,

on behalf of

Dr. Olle Terenius 

Academic Editor

PLOS ONE